# Early Effect of Supplementation with Essential Amino Acids on Cardiac Performance in Elderly Patients with Heart Failure and Sarcopenia

**DOI:** 10.3390/ijms26157533

**Published:** 2025-08-04

**Authors:** Giuseppe Armentaro, Velia Cassano, Pasquale Loiacono, Carlo Fuoco, Giandomenico Severini, Carlo Alberto Pastura, Alberto Panza, Marilisa Panza, Elisa Mazza, Sofia Miceli, Arturo Pujia, Tiziana Montalcini, Angela Sciacqua

**Affiliations:** 1Geriatrics Division, “Renato Dulbecco” University Hospital of Catanzaro, 88100 Catanzaro, Italy; giuseppearmentaro91@gmail.com (G.A.); sofy.miceli@libero.it (S.M.); 2Department of Medical and Surgical Sciences, University Magna Græcia of Catanzaro, 88100 Catanzaro, Italy; velia.cassano@unicz.it (V.C.); pa.loiacono@libero.it (P.L.); carlofuoco@gmail.com (C.F.); giandomenicoseverini@gmail.com (G.S.); carloalbertopastura@gmail.com (C.A.P.); albertopanza94@gmail.com (A.P.); marilisa.panza@studenti.unicz.it (M.P.); elisamazza@unicz.it (E.M.); pujia@unicz.it (A.P.); 3Research Center for the Prevention and Treatment of Metabolic Diseases, University Magna Græcia, 88100 Catanzaro, Italy; tmontalcini@unicz.it; 4Department of Clinical and Experimental Medicine, University Magna Græcia, 88100 Catanzaro, Italy

**Keywords:** heart failure, sarcopenia, oxidative stress, essential amino acids

## Abstract

The aim of the present observational study was to evaluate the early effect of free-form essential amino acid (EAA) supplementation on cardiac and muscular performance in elderly patients with chronic heart failure (HF) with reduced ejection fraction (HFrEF) and sarcopenia, as add-on to the optimized medical therapy (OMT) for HF. The present study included 60 elderly Caucasian patients suffering from HFrEF and sarcopenia. At the baseline and at follow-up, all patients underwent complete physical examination with the determination of the main anthropometric and hemodynamic parameters. After 6 months of supplementation with EAAs, we observed significant improvements in the parameters of sarcopenia. In addition, there was a significant improvement in glycol-metabolic parameters, and in inflammatory index as high sensitivity C-reactive protein (hs-CRP). In accordance with these results, significant decreases were observed in circulating levels of oxidative stress biomarkers Nox-2 (*p* < 0.001) and 8-Isoprostane (*p* < 0.001), and platelet aggregation biomarkers such as sP-Selectin (*p* < 0.001) and Gp-VI (*p* < 0.001). Of particular interest, after 6 months’ follow-up, there was a significant improvement in LVEF and global longitudinal strain (GLS). In conclusion, this study demonstrates that targeted nutritional intervention with EEAAs represents a viable therapeutic strategy for addressing the complex interplay between cardiac dysfunction and skeletal muscle wasting in elderly HF patients.

## 1. Introduction

Heart failure (HF) is a complex clinical syndrome characterized by typical signs and symptoms due to a structural or functional cardiac abnormality that results in inadequate cardiac output at rest or during exercise and/or elevated intraventricular filling pressures [1]. It mainly affects the elderly population, where the prevalence exceeds 10%, and is associated with several comorbidities that worsen quality of life and prognosis [1]. HF is characterized as a hypercatabolic condition, which involves all organs and is characterized by skeletal muscle protein degradation, causing skeletal muscle wasting [2]. Modifications in skeletal mass include a reduction in type I muscle fibres, a decrease in the oxidative capacity and cross-sectional area of type II muscle fibres, a reduction in mitochondrial volume within muscle fibres, a reduction in enzymes required for aerobic metabolism, and an increase in glycolytic enzymes, increasing the risk of sarcopenia [3].

Sarcopenia, a primary age-dependent syndrome, is defined as the progressive loss of skeletal muscle mass, strength, and function, and represents a major comorbidity in HF patients, contributing to reduced physical capacity, frailty, and worse clinical outcomes [4,5].

The pathophysiological mechanisms underlying sarcopenia in patients with HF involve complex interactions between chronic low-grade inflammation, disrupted anabolic–catabolic protein balance, and metabolic dysregulation [6]. Disrupted branched-chain amino acid (BCAA) catabolism is a prominent pathway in sarcopenia, which leads to BCAA accumulation and decreased muscle capacity.

Therefore, treatments that are qualified to prevent the development of sarcopenia in HF patients are needed. Dietary implementation with proteins or essential amino acids (EAAs) has been shown to increase skeletal muscle strength and mass quality in patients with chronic conditions such as metabolic syndrome, dementia, and HF [7].

Amino acids (AAs) are molecules containing amine group, carboxylic acid group a side chain. EAAs cannot be synthesized de novo by the organism; therefore, they must be provided by the organism’s diet [2].

Recent preclinical evidence suggests that EAA supplementation extends survival primarily through promotion of mitochondrial biogenesis, addressing one of the fundamental cellular energy deficits observed in both HF and sarcopenia [8].

The mechanistic rationale for EAA supplementation in HF with concurrent sarcopenia encompasses multiple pathways. BCAAs, including leucine, isoleucine, and valine, play significant roles in blood glucose regulation, protein synthesis, and insulin sensitivity [9]. Furthermore, BCAA preparations, which are essential for skeletal muscle formation, may be beneficial in HF, promoting postoperative wound healing and recovery from muscle fatigue after exercise, as well as improving muscle strength [10].

Clinical evidence supporting this therapeutic approach continues to accumulate. Collectively, these results suggest that manipulation of dietary AAs, and especially EAAs, is a potential adjuvant therapeutic strategy to treat patients with HF [8]. This emerging therapeutic paradigm represents a convergence of nutritional science and cardiovascular medicine, offering a potentially transformative approach to managing the complex interplay between cardiac dysfunction and skeletal muscle wasting.

Based on previous evidence in the literature, the aim of the present observational study was to evaluate the early effect of free-form EAA supplementation on cardiac and muscular performance in elderly patients with chronic HF with reduced ejection fraction (HFrEF) and sarcopenia, as add-on to the optimized medical therapy (OMT) for HF. In addition, we also evaluated the variation of laboratory parameters between baseline and follow-up, including biomarkers of oxidative stress (Nox-2 and 8-Isoprostane) and platelet aggregation (sP-Selectin and Gp-VI).

## 2. Results

The present study included 60 elderly Caucasian patients (mean age 80.1 ± 1.1 years; 47 men and 13 women) suffering from HFrEF and sarcopenia. Considering associated comorbidities, 93% of the patients presented arterial hypertension, 88% exhibited dyslipidaemia, 70% had type 2 diabetes mellitus (T2DM), and 63% had ischemic heart disease (Table 1).

At baseline, all patients were on optimal medical therapy for HFrEF at the maximum tolerated dose; all patients were in treatment with Sacubitril/Valsartan (ARNI) and sodium-glucose co-transporter 2 inhibitors (SGLT2i); 92% of patients were treated with beta-blockers and 38% with mineralocorticoid receptor antagonists (MRAs) (Table 1). All patients received supplementation of 5.5 g of a free-form EAA mixture twice daily for the entire study duration (6 months).

After 6 months of supplementation with EAAs, we observed significant improvements in the parameters of sarcopenia as demonstrated by increased handgrip strength (from 18.97 ± 5.33 to 21.48 ± 5.16, *p* < 0.001); gait speed (from 0.79 (0.72–0.86) to 0.92 (0.85–0.98), *p* < 0.001); SARC-F (from 6.42 ± 1.93 to 3.95 ± 2.34, *p* < 0.001); and SPPB (from 5.2 ± 1.7 to 6.8 ± 1.7, *p* < 0.001) (Table 2) (Figure 1).

On the other hand, a significant reduction was observed in NT-pro-BNP values (from 2496.0 (936.0–3687.0) to 2012.0 (280.0–2656.0) pg/mL, *p* < 0.001), and improvement was observed in clinical symptoms (Kansas City Cardiomyopathy Questionnaire—clinical score (KCCQ-CS) from 60.9 ± 1.4 to 64.6 ± 1.6 pt, *p* < 0.001). Moreover, in particular we observed a significant reduction in SBP (123.1 ± 9.4 vs. 119.9 ± 9.6 mmHg, *p* < 0.001) and DBP (71.0 ± 6.5 vs. 67.7 ± 5.7 mmHg, *p* < 0.001). In addition, there was a significant improvement in glycol-metabolic parameters as HOMA index (7.4 ± 2.7 vs. 5.6 ± 2.3, *p* < 0.001) and HbA1c (7.1 ± 1.1 vs. 6.6 ± 1.2, *p* < 0.001) and in inflammatory index as high sensitivity C-reactive protein (hs-CRP) (7.6 ± 0.4 vs. 6.8 ± 0.5 mg/dL, *p* < 0.001). In accordance with these results, a significant decrease was observed in circulating levels of oxidative stress biomarkers Nox-2 (0.73 ± 0.13 vs. 0.53 ± 0.10 nmol/L, *p* < 0.001) and 8-Isoprostane (from 73.5 ± 10.8 to 54.7 ± 9.0 pg/mL, *p* < 0.001), and platelet aggregation biomarkers such as sP-Selectin (from 119.3 ± 19.6 to 989.1 ± 15.7 ng/mL, *p* < 0.001) and Gp-VI (from 58.2 ± 12.0 to 46.3 ± 12.0 pg/mL, *p* < 0.001) (Table 3) (Figure 2).

Of particular interest, after six months’ follow-up, there was a significant improvement in LVEF (from 34.3 ± 1.5 to 35.80 ± 1.7%, *p* < 0.001) and GLS (−8.5 ± 1.07 vs. −11.9 ± 1.23, *p* < 0.001). In addition, there was an improvement in diastolic function with a reduction in left ventricular filling pressures as evidenced by a reduction in E/e’ ratio (from 16.1 ± 3.1 to 15.1 ± 3.0, *p* < 0.001).

Moreover, between baseline and follow-up, there was a significant improvement in right heart parameters such as RVOTp (2.2 ± 0.3 vs. 2.1 ± 0.2 cm, *p* < 0.001), s-PAP (40 ± 6.8 vs. 37 ± 5.1 mmHg, *p* < 0.001), TAPSE (17.1 ± 1.9 vs. 17.8 ± 1.9 mm, *p* < 0.001), and TAPSE/s-PAP ratio (0.43 ± 0.09 vs. 0.49 ± 0.10 mm/mmHg, *p* < 0.001) (Table 4).

### Linear Regression Analysis

The simple linear regression model demonstrated that ΔGLS was significantly correlated with ΔHOMA (r = 0.342, *p* = 0.003), Δhs-CRP (r = 0.324, *p* = 0.007), and ΔNox-2 (r = 0.168, *p* = 0.015). Variables reaching statistical significance were introduced in a stepwise multivariate linear regression model to identify the independent predictors of ΔGLS. This model shows that Δhs-CRP correlates 25.8% with ΔGLS, while ΔHOMA and ΔNox-2 correlate 13.6% and 4.3%, respectively, with the dependent variable, and the entire statistical model correlates 43.7% with ΔGLS (Table 5 and Table 6).

## 3. Discussion

The present study demonstrates that the addition of EEAAs to the therapy of HFrEF and sarcopenic patients significantly improves clinical and echocardiographic variables assessed by speckle-tracking echocardiography, parameters of sarcopenia, and markers of oxidative stress and platelet activation. The novelty of the present study is that we demonstrated the central role of EEAAs in improving cardiac function functional limitations, muscle strength, and gait speed in patients with HFrEF and sarcopenia.

In recent years, many studies have highlighted the importance of macro- and micronutrient deficiencies in HF and their contribution to the pathophysiology and progression of the cardiac dysfunction that may lead to a reduction in exercise tolerance and quality of life.

It is known that HF is associated with physiological changes, including skeletal muscle dysfunction, malnutrition, and reduction in exercise and functional capacity, especially in elderly patients [5].

Muscle trophism results from a balance between anabolic stimuli (insulin, exercise, amino acids, IGFs, testosterone, adrenaline, GH) and catabolic stimuli (cortisol, catecholamines, glucagon, cytokines, intense exercise) [4].

Insufficient dietary protein intake represents a strong predictor of developing sarcopenia in patients with HF [11]. EAA supplementation plays an important role in this setting.

Data obtained from the present study demonstrated that after 6 months of oral EAA supplementation, there was a significant improvement across several domains. In addition, the metabolic profile also shows substantial improvements, with a reduction of HOMA, indicating enhanced insulin sensitivity.

Previous preclinical studies have already demonstrated that EAAs act through the mTOR-mediated pathway on reducing insulin resistance through insulin-independent intracellular glucose transport [12].

Moreover, the significant reduction in inflammatory markers and oxidative stress biomarkers (Nox-2 and 8-isoprostane) indicates attenuation of pathological inflammatory cascades, also confirmed by the reduction in circulating levels of hs-CRP. These improvements are mechanistically important, as oxidative stress and inflammation are key drivers of HF progression and myocardial dysfunction. Chronic inflammation is a key driver of muscle protein breakdown through activation of ubiquitin-proteasome pathways and inhibition of muscle protein synthesis [13]. The observed inflammatory marker improvements may therefore represent both a direct effect of the intervention and a contributory mechanism to muscle preservation and recovery.

After 6 months of EAA intake, we also observed a modest but statistically significant improvement in LV function. Most notably, GLS improvement (from −8.5% to −11.9%), represents enhanced myocardial contractility. GLS is an important parameter reflecting LV deformation in the longitudinal direction, and it is highly specific and earlier in detecting systolic dysfunction compared to LVEF. As evidenced by existing data in the literature, EAA supplementation allows improvement in echocardiographic and haemodynamic parameters. A study by Lombardi et al. had already shown that supplementation with EEAAs for 3 months improved functional abilities in patients with HF; however, in this study it was conducted on a small sample size compared to ours (13 vs. 60 patients), with a shorter follow-up (3 vs. 6 months). Furthermore, none of the patients were taking SGLT2i or ARNI, unlike in our study where 100% of the patients were taking these drug classes [14].

In addition, obtained data from the present study reveal particularly compelling evidence regarding the management of sarcopenia in elderly HF patients, with the introduction of EAAs representing a significant therapeutic advancement. Moreover, we observed a significant improvement in sarcopenia-related parameters such as SARC-F scores, handgrip, SPPB scores, and gait speed.

As a probable consequence of the improvement in cardiac function and functional abilities, there was an important improvement in the clinical symptoms detected with the KCCQ-CS and a significant reduction in NT-proBNP levels.

The concurrent improvement in sarcopenia-related parameters alongside the HOMA index, suggests that EAA supplementation provides multisystem benefits. This metabolic improvement is mechanistically important, as insulin resistance and muscle wasting are interconnected through shared pathways involving mTOR signalling and protein synthesis regulation. Enhanced insulin sensitivity likely contributes to improved muscle anabolism and functional recovery [12].

These findings suggest significant reversal of frailty and sarcopenia, which are critical prognostic factors in elderly patients in HF. As demonstrated by previous studies conducted on patients with sarcopenia, EAAs initiate the stimulation of muscle protein synthesis, which positively affects muscle strength measured by handgrip [15]. However, the study conducted by Kuczmarski et al. enrolled patients of a different ethnicity and without sarcopenia and HF, unlike our study, which enrolled Caucasian, elderly patients with these two comorbidities.

Another fundamental aspect to consider in these patients is poor exercise tolerance. Oral EAA supplementation, in combination with standard pharmacological therapy, appears to increase exercise capacity by improving circulatory function, muscle oxygen consumption, and aerobic energy production in elderly patients with chronic HF [16].

Although the previous study demonstrates a significant improvement in functional abilities and oxygen consumption in patients with HF, unlike our study, the patients were not treated with Sacubitril/Valsartan and SGLT2i, as these two drug classes were subsequently indicated for the treatment of HFrEF. Furthermore, the benefits were studied for only 30 days, while in our study the follow-up was 6 months.

The relationship between sarcopenia improvement and cardiac functional enhancement (GLS improvement) suggests bidirectional benefits. Improved muscle mass and function may reduce peripheral vascular resistance and improve overall CV efficiency. Conversely, improved cardiac function facilitates better tissue perfusion and nutrient delivery, supporting muscle anabolism.

The present study has some limitations when interpreting the results. First, one of the main limitation is the small sample size of patients and the lack of control group. Another limitation is that the present study includes enrolment of Caucasian patients only, so the application of these data to patients of other ethnicities is not clear. The future research direction would be to expand the study population and enrol not only Caucasian patients to see the effect of EAAs in patients of different ethnicities.

In conclusion, this study demonstrates that targeted nutritional intervention with EEAAs represents a viable therapeutic strategy for addressing the complex interplay between cardiac dysfunction and skeletal muscle wasting in elderly heart failure patients, offering hope for improved functional outcomes and quality of life in this vulnerable population. In particular, this study shows that supplementation with EEAAs in elderly patients with HFrEF and sarcopenia, with several comorbidities, is associated with significant improvements in the metabolic and inflammatory profile and in biomarkers of oxidative stress after just 6 months of treatment. Furthermore, the improvement in these parameters is associated with improved cardiac performance as assessed by GLS. Therefore, supplementation with EEAAs could represent an additional therapeutic strategy for elderly patients with HFrEF and sarcopenia who are already undergoing therapy.

## 4. Materials and Methods

### 4.1. Design and Participants

In the present single-centre observational study, we enrolled 60 outpatients (mean age 80.1 ± 1.1, 47 males and 13 females) afferent to the Geriatrics Unit of the Renato Dulbecco University Hospital.

The protocol was approved by the University Ethics Committee (2022.384), and written informed consent was obtained from all participants to the “Magna Graecia Evaluation of Comorbidities in Patients with Heart Failure (MAGIC-HF)” study (ClinicalTrials.gov identifier: NCT05915364) and by the local Ethics Committee of Calabria Region, Italy (Catanzaro, Italy, document n. 263–23 July 2020). This study met the standards of good clinical practice (GCP) and the principles of the Declaration of Helsinki.

All patients presented diagnosis of HFrEF and sarcopenia according to ESC guidelines and were in optimal pharmacological treatment, at the enrolment time.

Exclusion criteria wer: chronic kidney disease stage IV K-DOQI (eGFR < 30 mL/min/1.73 m^2^, CKD-EPI), severe hepatic impairment (Child-Pugh Class C), history of angioedema, and previous diagnosis of dementia or serious psychiatric disorders.

### 4.2. Study Procedures

At the baseline, all patients underwent complete physical examination with determination of the main anthropometric and hemodynamic parameters. Relevant comorbidities and the type of drug therapies were also recorded. Evaluation of the NYHA functional class was carried out as suggested by current guidelines [1].

All patients were screened for possible and probable presence of sarcopenia (SARC-F questionnaire) [11] and evaluation of muscle strength with the handgrip test (HGS) [12]. The SARC-F is a 5-item self-reported screening questionnaire used to assess sarcopenia risk. It includes five components: strength, assistance walking, rise from a chair, climb stairs, and falls. SARC-F scale scores range from 0 to 10. The Kansas City Cardiomyopathy Questionnaire (KCCQ-CS) is a valid, reliable, and sensitive test for assessing the health status and symptoms of patients with CHF, which is administered to the patient in the form of a questionnaire [13]. Moreover, we analysed the condition of sarcopenia with BIA and evaluated the severity of the condition through short physical performance battery (SPPB) [5]. SPPB is a composite test used to assess physical function. It includes assessment of gait speed, a balance test, and chair stand test. Gait speed is evaluated over a four-meter distance. Static balance is evaluated through three tests: single-leg stance, semi-tandem stance, and tandem stance; each of them lasts 10 s.

The blood pressure (BP) measurements were acquired in the left arm of patients in sitting position using a semi-automatic sphygmomanometer (OMRON, M7 Intelli IT) after five min of rest. Subjects with a clinic systolic BP (SBP)  >  140 mmHg and/or diastolic BP (DBP)  >  90 mmHg were defined as hypertensive. Pulse pressure (PP) values were acquired as the difference between SBP and DBP measurements [14].

### 4.3. Laboratory Tests

After at least 12 h fasting, the laboratory measurements were performed. The glucose oxidation method (Beckman Glucose Analyzer II; Beckman Instruments, Milan, Italy) was utilized to measure plasma glucose, and a chemiluminescence-based assay (Roche Diagnostics) was used for plasma insulin determination. Insulin sensitivity was determined with the metabolic homeostasis method (Homeostasis Model Assessment of Insulin Resistance, HOMA). An enzymatic method (Roche Di-agnostics GmbH, Mannheim, Germany) was used to detect total, low, and high-density lipoprotein (LDL, HDL) cholesterol and triglyceride concentrations. Serum creatinine was determined using a Roche Creatinine Plus assay (Hoffman-La Roche, Basel, Switzerland) on a clinical chemistry analyser (Roche/Hitachi Modular Analytics System, P Module). Renal function was evaluated by calculating the estimate glomerular filtration rate (e-GFR) using the CDK-EPI equation [15].

N-terminal pro-brain natriuretic peptide (NT-proBNP) levels were assessed by an enzyme-linked immunosorbent assay (Elecsys proBNP assay, Roche Diagnostics). Serum sodium and potassium levels were obtained by indirect potentiometry (Cobas, Roche) and high sensitive C-reactive protein (hs-CRP) by an automated instrument (Cardio-Phase1hsCRP), Milan, Italy).

The quantification of oxidative stress (8-isoprostane and Nox-2) and platelet activation (Glycoprotein-VI and sP-selectin) biomarkers was conducted as previously described [16,17].

### 4.4. Echocardiographic Parameters

Echocardiographic recordings were performed using a VIVID E-95 ultrasound system (GE Technologies, Milwaukee, WI, USA) with a 2.5 MHz transducer. All patients were examined at rest and in the left lateral decubitus position. Measurements were obtained according to the recommendations of the American Society of Echocardiography [18]. To minimize measurement errors, the echocardiographic examinations were carried out by the same expert operator, who was not aware of the patient’s clinical data; the values considered represent the average of at least three measurements. Left ventricular mass (LVM) was calculated using the formula proposed by Devereux and corrected for body surface area (BSA), to derive the LVM index (LVMI) [19]. Among the parameters of left ventricular global systolic function, left ventricular ejection fraction (LVEF) and cardiac index (CI) were evaluated [18]. LVEF was calculated by the Simpson biplane method. Right ventricular systolic parameters were also measured, by estimating the systolic pulmonary arterial pressure (S-PAP) [20].

The diameter of the right ventricular outflow tract (RVOT) and the right atrium area (RAA) were obtained according to ASE recommendations [18]. The movement of the tricuspid annulus was recorded at the free wall of the RV for the tricuspid annular plane systolic excursion (TAPSE), which expresses the right longitudinal function.

A 2D speckle-tracking analysis was performed using vendor-specific 2D speckle-tracking software (EchoPAC PC, version 113.0.5, GE Healthcare, Horten, Norway). Manual tracings of the endocardial border during end-systole in three apical views were performed to evaluate global longitudinal strain (GLS) [21].

### 4.5. Statistical Analysis

Continuous variables were expressed as mean ± standard deviation (SD) (normally distributed data) or as median and interquartile range (IQR) (non-normally distributed data). Categorical data were expressed as percentages. Longitudinal changes in key variables at follow-up were analysed with the *t*-test or Wilcoxon’s test for paired data. For categorical data, comparison was conducted using chi-square test. A simple linear regression analysis was performed to assess the correlation between GLS index, expressed as Δ of variation between baseline and follow-up (ΔT0–6), and the variation of metabolic, inflammatory, oxidative stress, platelet activation, and renal function covariates, also expressed as ΔT0–6. Variables that reached statistical significance were entered into a stepwise multivariate linear regression model to evaluate the magnitude of their individual effect on ΔGLS.

## Figures and Tables

**Figure 1 ijms-26-07533-f001:**
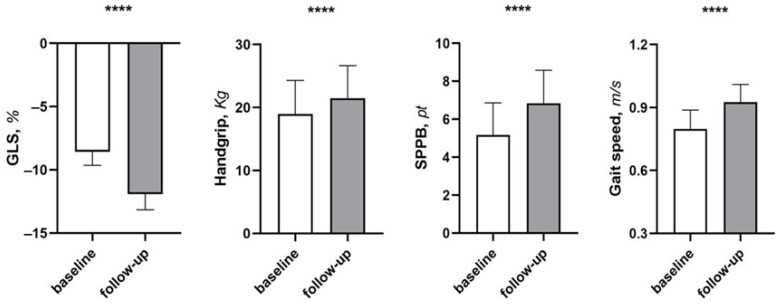
Changes in echocardiographic parameters and sarcopenia index between baseline and follow-up. Data are mean ± SD. **** *p* < 0.0001 vs. baseline. **Abbreviations.** GLS: global longitudinal strain; SPPB: short performance physical battery.

**Figure 2 ijms-26-07533-f002:**
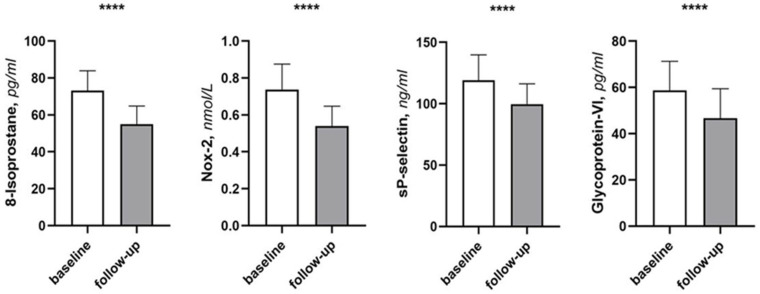
Changes in oxidative stress biomarkers and platelet activation biomarkers between baseline and follow-up. Data are mean ± SD. **** *p* < 0.0001 vs. baseline. **Abbreviations:** Nox-2: NAPDH Oxidase 2.

**Table 1 ijms-26-07533-t001:** Clinical characteristics, comorbidity drug therapies of the study population at baseline.

	Whole Population (*n* = 60)
**Gender (Male)**, *n* (%)	47 (78)
**Gender (Female)**, *n* (%)	13 (22)
**Mean age**, *years*	80.12 ± 1.1
**IHD**, *n* (%)	38 (63)
**Heart valve disease**, *n* (%)	21 (35)
**AF**, *n* (%)	25 (42)
**T2DM**, *n* (%)	42 (70)
**AH**, *n* (%)	56 (93)
**COPD**, *n* (%)	23 (38)
**Dyslipidaemia**, *n* (%)	53 (88)
**Smokers**, *n* (%)	23 (38)
**ARNI**, *n* (%)	60 (100)
**SGLT2i**, *n* (%)	60 (100)
**MRAs**, *n* (%)	23 (38)
**Statins**, *n* (%)	50 (83)
**β-blockers**, *n* (%)	55 (92)
**ICD/CRT-D**, *n* (%)	30 (50)
**OACs**, *n* (%)	20 (33)
**Antiplatelet drug**, *n* (%)	28 (47)
**Diuretics**, *n* (%)	55 (92)
**OADs**, *n* (%)	34 (57)

**Abbreviations.** IHD: ischaemic heart disease; AF: atrial fibrillation; T2DM: type 2 diabetes mellitus; AH: arterial hypertension; COPD: chronic obstructive pulmonary disease; ARNI: angiotensin receptor neprilysin inhibitor; SGLT2i: sodium glucose cotransporter 2 inhibitors; MRAs: mineralocorticoid receptor antagonists; ICD: implantable cardioverter defibrillator; CRT-D: Cardiac resynchronization therapy with or without a defibrillator; OACs: oral anticoagulants; OADs: oral antidiabetic drugs.

**Table 2 ijms-26-07533-t002:** Clinical characteristics of the study population at baseline and follow-up.

Clinical Characteristics	Baseline	Follow-Up	*p* Value
**KCCQ-CS**, pt	60.9 ± 1.4	64.6 ± 1.6	<0.001
**BMI**, Kg/m^2^	33.1 ± 4.0	32.2 ± 3.8	<0.001
**SBP**, mmHg	123.1 ± 9.4	119.9 ± 9.6	<0.001
**DBP**, mmHg	71.0 ± 6.5	67.7 ± 5.7	<0.001
**HR**, beats/min	81.1 ± 9.7	74.9 ± 7.0	<0.001
**HOMA**, index	7.4 ± 2.7	5.6 ± 2.3	<0.001
**SARC-F**, pt	6.42 ± 1.93	3.95 ± 2.34	<0.001
**HANDGRIP**, Kg	18.97 ± 5.33	21.48 ± 5.16	<0.001
**SPPB**, pt	5.2 ±1.7	6.8 ±1.7	<0.001
**Gait speed**, m/s	0.79 (0.72–0.86)	0.92 (0.85–0.98)	<0.001

**Abbreviations.** KCCQ-CS: Kansas City Cardiomyopathy Questionnaire; BMI: body mass index; SBP: systolic blood pressure; DBP: diastolic blood pressure; HR: heart rate; HOMA: homeostatic model assessment; SARC-F: strength, assistance with walking, rise from chair, climb stairs, falls; SPPB: short physical performance battery.

**Table 3 ijms-26-07533-t003:** Biochemical parameters of the study population at baseline and follow-up.

Biochemical Parameters	Baseline	Follow-Up	*p*-Value
**Hb**, g/dL	11.7 ± 1.12	12.8 ± 1.66	<0.001
**Na**, mmol/L	139.8 ± 1.7	139.1 ± 1.2	<0.001
**K**, mmol/L	4.4 ± 0.3	4.5 ± 0.45	<0.001
**LDL**, mg/dL	85 ± 33.7	80 ± 29.5	<0.001
**HDL**, mg/dL	39 ± 9	40 ± 8	<0.001
**Triglycerides**, mg/dL	183.8 ± 60	157 ± 45	<0.001
**Creatinine**, mg/dL	1.15 ± 0.35	1.07 ± 0.31	<0.001
**e-GFR**, mL/min/1.73 m^2^	64 ± 19	86 ± 24	<0.001
**HbA1c**, %	7.1 ± 1.2	6.6 ± 1.2	<0.001
**NT-proBNP**, pg/mL	2496.0 (936.0–3687.0)	2012.0 (280.0–2656.0)	<0.001
**Uric acid**, mg/dL	6.7 ± 0.54	6.11 ± 0.96	<0.001
**hs-CRP**, g/dL	7.6 ± 0.4	6.8 ± 0.5	<0.001
**Nox-2**, nmol/L	0.73 ± 0.13	0.53 ± 0.10	<0.001
**8-isoprostane**, pg/mL	73 ± 10	54 ± 9	<0.001
**Gp-VI**, pg/mL	58 ± 12	46 ± 12	<0.001
**Sp-selectin**, ng/mL	119 ± 20	99 ± 16	<0.001

**Abbreviations.** Hb: Haemoglobin g/dL; Na: sodium; K: potassium; LDL: low density lipoprotein; HDL: high density lipoprotein; eGFR: glomerular filtration rate; HbA1c: glycated haemoglobin; hs-CRP: high sensitivity C-reactive protein; NOX-2: NAPDH oxidase; Gp-VI: glycoprotein-6.

**Table 4 ijms-26-07533-t004:** Echocardiographic parameters of the study population at baseline and follow-up.

Echocardiographic Parameters	Baseline	Follow-Up	*p* Value
**LAVi**, mL/m^2^	46.5 ± 13.1	43.2 ± 9.8	<0.001
**LVEDV/BSA**, mL/m^2^	90.1 ± 9.1	82.5 ± 9.9	<0.001
**LVESV/BSA**, mL/m^2^	59.1 ± 6.1	52.9 ± 6.6	<0.001
**LVEF**, %	34.3 ± 1.5	35.8 ± 1.7	<0.001
**E/A**	0.62 ± 0.10	0.68 ± 0.12	<0.001
**E/e’**	16.1 ± 3.1	15.1 ± 3.0	<0.001
**GLS**, %	−8.5 ± 1.07	−11.9 ± 1.23	<0.001
**RVOT**, cm	2.2 ± 0.3	2.1 ± 0.2	<0.001
**Right atrial area**, cm^2^	19 ± 2.7	17 ± 1.6	<0.001
**TAPSE**, mm	17.1 ± 1.9	17.8 ± 1.9	<0.001
**PAPS**, mmHg	40 ± 6.8	37 ± 5.1	<0.001
**TAPSE/PAPs**, mm/mmHg	0.43 ± 0.09	0.49 ± 0.10	<0.001
**IVC**, mm	18.8 ± 1.65	17.4 ± 1.23	<0.001

**Abbreviations.** LAVi: left atrial volume index; LVEDV: left ventricular end diastolic volume; BSA: body surface area; LVESV: left ventricular end systolic volume; LVEF: left ventricular ejection fraction; GLS: global longitudinal strain; RVOT: right ventricular outflow tract; TAPSE: tricuspid annulus plane systolic excursion; PAPS: systolic pulmonary artery pressure; IVC: inferior vena cava.

**Table 5 ijms-26-07533-t005:** Simple linear regression model between Δ of GLS as dependent variable and Δ of different covariates in the study population.

	ΔGLS	
	r	*p*
**ΔeGFR**, mL/min/1.73 m^2^	−0.066	0.574
**ΔHOMA**	**0.342**	**0.003**
**Δ8-Isoprostane**, pg/mL	0.095	0.362
**ΔNox-2**, nmol/L	**0.168**	**0.015**
**ΔUric acid**, mg/dL	0.091	0.388
**Δhs-CRP**, mg/L	**0.324**	**0.007**

**Abbreviations.** GLS: global longitudinal strain; eGFR: glomerular filtration rate; HOMA: homeostatic model assessment; Nox-2: NAPDH Oxidase 2; hs-CRP: highly sensitive C-reactive protein.

**Table 6 ijms-26-07533-t006:** Stepwise multivariate regression model on ΔGLS as the dependent variable in the study population.

ΔGLS	R^2^ Partial	R^2^ Total	*p*
**Δhs-CRP**, mg/L	25.8%	25.8%	<0.0001
**ΔHOMA**	13.6%	39.4%	0.001
**ΔNox-2**, nmol/L	4.3%	43.7%	0.043

**Abbreviations:** GLS, global longitudinal strain; HOMA: homeostatic model assessment; Nox-2: NAPDH Oxidase 2; hs-CRP: highly sensitive c-reactive protein.

## Data Availability

The original contributions presented in this study are included in the article. Further inquiries can be directed to the corresponding author.

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
