# Peer review of "Early Effect of Supplementation with Essential Amino Acids on Cardiac Performance in Elderly Patients with Heart Failure and Sarcopenia"

_ijms, 2025, doi:10.3390/ijms26157533_

Round 1
Reviewer 1 Report
Comments and Suggestions for Authors
Early effect of supplementation with essential amino acids on 2
cardiac performance in elderly patients with heart failure and 3
sarcopenia
The study discussing effects of BAA supplementation in elderly patients suffering of heart failure and sarcopenia.
The study only included experimental group no control group, the effect of treatment based on cases only, effect after 6-month duration without tight control subjected to similar conditions is a bit challenging and could encompass a lot of confounding. Could you add relevant control or at least justify your experimental design.
Even the studies cases are variable in their co-morbidity or drugs administrated, smoking, gender and age – data are not stratified to reflect any association.
Method: the gait speed is not mentioned how it is evaluated and what measures taken, the Short physical performance baĴery (SPPB) has no description, same for SARC-F and KCCQ-CS questionnaires-in addition to reference you can briefly describe it please.
Thanks
Author Response
Reviewer 1
Early effect of supplementation with essential amino acids on 2 cardiac performance in elderly patients with heart failure and 3 sarcopenia. The study discussing effects of BAA supplementation in elderly patients suffering of heart failure and sarcopenia.
Q: The study only included experimental group no control group, the effect of treatment based on cases only, effect after 6-month duration without tight control subjected to similar conditions is a bit challenging and could encompass a lot of confounding. Could you add relevant control or at least justify your experimental design.
R: Thank you for this observation, which allows us to clarify a fundamental aspect of the manuscript. We agree with the reviewer, and we have included the absence of a control group in the limitations of the study, see discussion section. However, we selected only this group because our purpose was to evaluate the possible effect of essential amino acid supplementation in a cohort of patients with chronic HFrEF and sarcopenia, already on top of therapy, not only on sarcopenia parameters, but also on echocardiographic parameters, circulating NTproBNP levels, and biomarkers of oxidative stress and platelet activation. However, in the future, we may increase the sample size and compare it with a control group for a longer follow-up.
Even the studies cases are variable in their co-morbidity or drugs administrated, smoking, gender and age – data are not stratified to reflect any association.
R: Thank you for this suggestion, however, given the small sample size (60 patients), it was not possible to perform a subgroup analysis with good statistical significance. However, in the future, with a larger sample size and longer follow-up, we could correct the analysis for these variables.
Method: the gait speed is not mentioned how it is evaluated and what measures taken, the Short physical performance baĴery (SPPB) has no description, same for SARC-F and KCCQ-CS questionnaires-in addition to reference you can briefly describe it please.
R: Thank you for this suggestion, which allows us to improve a fundamental methodological aspect of the manuscript. We have made the requested changes; see the Materials and Methods section.
Reviewer 1
Early effect of supplementation with essential amino acids on 2 cardiac performance in elderly patients with heart failure and 3 sarcopenia. The study discussing effects of BAA supplementation in elderly patients suffering of heart failure and sarcopenia.
Q: The study only included experimental group no control group, the effect of treatment based on cases only, effect after 6-month duration without tight control subjected to similar conditions is a bit challenging and could encompass a lot of confounding. Could you add relevant control or at least justify your experimental design.
R: Thank you for this observation, which allows us to clarify a fundamental aspect of the manuscript. We agree with the reviewer, and we have included the absence of a control group in the limitations of the study, see discussion section. However, we selected only this group because our purpose was to evaluate the possible effect of essential amino acid supplementation in a cohort of patients with chronic HFrEF and sarcopenia, already on top of therapy, not only on sarcopenia parameters, but also on echocardiographic parameters, circulating NTproBNP levels, and biomarkers of oxidative stress and platelet activation. However, in the future, we may increase the sample size and compare it with a control group for a longer follow-up.
Even the studies cases are variable in their co-morbidity or drugs administrated, smoking, gender and age – data are not stratified to reflect any association.
R: Thank you for this suggestion, however, given the small sample size (60 patients), it was not possible to perform a subgroup analysis with good statistical significance. However, in the future, with a larger sample size and longer follow-up, we could correct the analysis for these variables.
Method: the gait speed is not mentioned how it is evaluated and what measures taken, the Short physical performance baĴery (SPPB) has no description, same for SARC-F and KCCQ-CS questionnaires-in addition to reference you can briefly describe it please.
R: Thank you for this suggestion, which allows us to improve a fundamental methodological aspect of the manuscript. We have made the requested changes; see the Materials and Methods section.
Reviewer 2 Report
Comments and Suggestions for Authors
Good morning,
First of all, thank you for the opportunity to review your article. It is always very enriching to evaluate the work of other researchers, as it allows us to learn and gain a deeper understanding of certain aspects of medicine.
Secondly, I would like to share some suggestions that you may find useful to improve your manuscript:
a) The graphical abstract should not be included within the main document. If necessary, it could be placed in the annexes section.
b) Before the results section, you should include a methodology section describing details such as the study population, sample size calculation, statistical analysis to be performed, and approval by the ethics committee. This section should always follow the introduction. Please also include the inclusion and exclusion criteria for your study.
c) It would be advisable to add a separate section—outside the discussion—that not only addresses the limitations of your study but also outlines future research directions that could help overcome them.
d) The conclusions should be clarified further to ensure that the main findings and implications are clearly stated.
e) The discussion could be improved not only by the references you have already included, but also by integrating a comparison between your findings and those of the studies you mention, highlighting points of agreement or divergence.
Once again, thank you for the opportunity, and I wish you the best as you continue to develop and complete your research.
Author Response
Reviewer 2
Good morning,
First of all, thank you for the opportunity to review your article. It is always very enriching to evaluate the work of other researchers, as it allows us to learn and gain a deeper understanding of certain aspects of medicine.
Secondly, I would like to share some suggestions that you may find useful to improve your manuscript:
- a) The graphical abstract should not be included within the main document. If necessary, it could be placed in the annexes section.
R: Thank you for the comment, we excluded the graphical abstract from the main document.
- b) Before the results section, you should include a methodology section describing details such as the study population, sample size calculation, statistical analysis to be performed, and approval by the ethics committee. This section should always follow the introduction. Please also include the inclusion and exclusion criteria for your study.
- We thank the reviewer for this observation. However, the ‘Materials and Methods’ section is included at the end of the manuscript, after the Discussion section, in accordance with the journal's editorial guidelines. Inclusion and exclusion criteria are in Materials and Methods section, Design and Participants paragraph. Regarding power calculation/sample size, statistically significant improvements in the main study variables were observed in our study population, and no worsening or non-statistically significant results were found that would indicate possible reduced study power and inadequate sample size; for these reasons, we did not perform power calculation and sample size calculation.
- c) It would be advisable to add a separate section—outside the discussion—that not only addresses the limitations of your study but also outlines future research directions that could help overcome them.
- Thanks for this observation. At the end of Discussion section, after study limitation, we added outlines future research direction.
- d) The conclusions should be clarified further to ensure that the main findings and implications are clearly stated.
R.Thank you for this suggestion. We have made the requested changes in the manuscript.
“In particular, this study shows that supplementation with EEAAs in elderly patients with HFrEF and sarcopenia, with several comorbidities, is associated with significant improvements in the metabolic and inflammatory profile and in biomarkers of oxidative stress after just 6 months of treatment. Furthermore, the improvement in these parameters is associated with improved cardiac performance as assessed by GLS. Therefore, supplementation with EEAAs could represent an additional therapeutic strategy for elderly patients with HFrEF and sarcopenia who are already on top of therapy”.
- e) The discussion could be improved not only by the references you have already included, but also by integrating a comparison between your findings and those of the studies you mention, highlighting points of agreement or divergence.
- We thank the reviewer for the comment. In the Discussion section, we compared our data with mentioned study.
Once again, thank you for the opportunity, and I wish you the best as you continue to develop and complete your research.
Round 2
Reviewer 1 Report
Comments and Suggestions for Authors
Thanks for adding the relevant methods section and describing the study limitations, you could include small sample size as a limitation too.
Reviewer 2 Report
Comments and Suggestions for Authors
Thanks for your considerations